# HYBRID DISTILLATION: CONNECTING MASKED AUTOENCODERS WITH CONTRASTIVE LEARNERS

**Bowen Shi**[1]  **Xiaopeng Zhang**[2*]  **Yaoming Wang**[1]  **Jin Li**[1]
**Wenrui Dai**[1]  **Junni Zou**[1]  **Hongkai Xiong**[1]  **Qi Tian**[2]
[1]Shanghai Jiao Tong University  [2]Huawei Inc.
`{sjtu_shibowen, wang_yaoming, deserve_lj, daiwenrui, zoujunni,`
`xionghongkai}@sjtu.edu.cn; zxphistory@gmail.com, tian.qi1@huawei.com`

## ABSTRACT

As two prominent strategies for representation learning, Contrastive Learning (CL) and Masked Image Modeling (MIM) have witnessed significant progress. Previous studies have demonstrated the advantages of each approach in specific scenarios. CL, resembling supervised pre-training, excels at capturing longer-range global patterns and enhancing feature discrimination, while MIM is adept at introducing local and diverse attention across transformer layers. Considering the respective strengths, previous studies utilize feature distillation to inherit both discrimination and diversity. In this paper, we thoroughly examine previous feature distillation methods and observe that the increase in diversity mainly stems from asymmetric designs, which may in turn compromise the discrimination ability. To strike a balance between the two properties, we propose a simple yet effective strategy termed **Hybrid Distill**, which leverages both the CL and MIM teachers to jointly guide the student model. Hybrid Distill emulates the token relations of the MIM teacher at intermediate layers for diversity, while simultaneously distilling the final features of the CL teacher to enhance discrimination. A progressive redundant token masking strategy is employed to reduce the expenses associated with distillation and aid in preventing the model from converging to local optima. Experimental results demonstrate that Hybrid Distill achieves superior performance on various benchmark datasets. The code is available at `https://github.com/lygsbw/hybriddistill`.

## 1 INTRODUCTION

Self-supervised pre-training has recently emerged as a promising alternative to supervised image classification (He et al., 2016; Dosovitskiy et al., 2020), particularly with Contrastive Learning (CL) and Masked Image Modeling (MIM). The former one, typical representatives are MoCo (He et al., 2020) and SimCLR (Chen et al., 2020a), learns invariant representation for positive views via different augmentations of the same image. Furthermore, CLIP (Radford et al., 2021) extends CL in a multi-modal manner by pairing the given image with its corresponding text description. While the latter, including MAE (He et al., 2022) and SimMIM (Xie et al., 2022b), aims to reconstruct the masked image patches and has become mainstream due to its efficiency brought by mask operations.

The different pre-training paradigms of CL and MIM advance a series of studies (Xie et al., 2022a; Park et al., 2023; Wang et al., 2023) that aim at understanding their intrinsic properties. These studies point out that CL behaves more similarly to supervised pre-training, *i.e.*, it provides models with longer-range global patterns focusing on object shape, particularly in the last few layers (Park et al., 2023), which enables feature representation with better **discrimination** that are beneficial for recognition (Huang et al., 2022). However, as shown in Fig. 1, it causes self-attention in the last few layers to collapse into homogeneity, with attention distances located within a small distance range. In contrast, MIM brings both local and global attention and evenly distributed representations across all layers (Xie et al., 2022a; Park et al., 2023), and this attention **diversity** contributes to its better generalization on downstream fine-tuning specifically for dense-level tasks (Park et al., 2023). Nevertheless, MIM underperforms in linear probing, mainly due to its lack of discrimination ability.

---

*Corresponding author. This work was done when Bowen Shi interned at Huawei Inc.

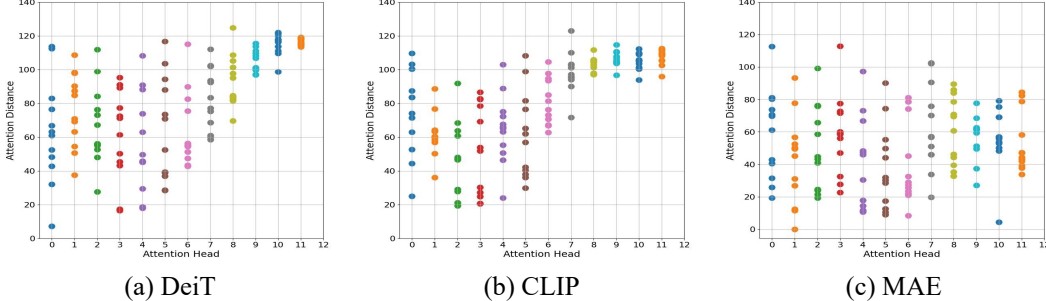

Figure 1: Average attention of (a) DeiT, (b) CLIP and (c) MAE pre-trained models.

Previous methods (Wei et al., 2022b; Fang et al., 2022; Liu et al., 2022; Wei et al., 2022a; Peng et al., 2022) propose to utilize feature distillation to ensure both discrimination and diversity. Among them, dBOT (Liu et al., 2022) replaces the reconstructing objective of MAE with the feature maps of different pre-trained teachers. It finds that feature distillation can bring diverse attention regardless of the teacher model, and after multi-stage distillation, the performance is comparable across different teachers even with the randomly initialized ones. Also observing that distillation benefits diversity, FD (Wei et al., 2022b) directly distills feature maps from supervised/CL teachers to relieve the attention collapse and achieves considerable downstream performance gains. Although interesting and important, we argue that their findings are incomplete.

This paper re-examines these findings and reconsiders the importance of diversity and discrimination. We reveal the following observations: (i) **The increase in diversity derives from the asymmetric architecture designs, rather than feature distillation itself.** (Section 2.2) After removing the asymmetric attention in Wei et al. (2022b) and encoder-decoder designs in Liu et al. (2022), we observe a negligible increase (or even a decrease) in attention diversity. (ii) **The asymmetric decoder de facto harm the discrimination over the encoder side, for it migrates the semantic information of the teacher model.** (Section 2.3) Due to the decomposition of the encoding and decoding functions, the student encoder tends to summarize more general information, thus gradually losing semantics obtained from teachers and yielding similar results after multi-stage distillation (Liu et al., 2022). (iii) **Mask reconstruction of high-level semantics does not help improve diversity.** (Section 2.4) Reconstructing high-level information (Peng et al., 2022; Fang et al., 2022; Wei et al., 2022a) is similar to direct feature distillation and lacks the diversity found in MIM, which implies that the attention diversity of MIM mainly comes from low-level reconstruction objectives.

Based on the above observations, we argue that uni-model distillation is limited for transferring both diversity and discrimination to the student side. To solve this issue, we propose a simple yet effective feature distillation method, termed as **Hybrid Distill**, to simultaneously inherit these two properties via distilling knowledge from both the supervised/CL and MIM teachers. Hybrid Distill makes careful designs for the distilling target and location. Specifically, we find that **the relational modeling ability of MIM is crucial for preserving token diversity, while the feature maps of supervised/CL teachers are beneficial for discrimination**. Accordingly, we set the token relations of the MIM teacher and the feature maps of the supervised/CL teacher as the distilling objectives. The token relations are distilled in layers preceding the final layer where attention collapse tends to occur, while the feature maps are distilled in the final layer to preserve semantics. Additionally, Hybrid Distill utilizes a progressive redundant token masking strategy to reduce distilling costs and prevent falling into local optima. Experiment results show that the distilling strategy works surprisingly well even when using MAE and CLIP teachers, *i.e.*, MAE pretrained with only 1.28M ImageNet images also boosts the large-scale (400M) pretrained CLIP teacher on different downstream tasks.

In a nutshell, this paper makes the following contributions:

• We re-examine the findings of previous feature distilling methods and point out that their increasing diversity mainly arises from the use of asymmetric designs, while these designs may in turn compromise the discrimination.

• We further propose a Hybrid Distill framework that utilizes both supervised/CL and MIM teachers to provide the student with higher-quality discrimination and diversity. Distilling targets and locations are carefully designed in Hybrid Distill to fully exploit the strengths of both teachers.

• We conduct property analysis to demonstrate that the representations exhibit both discrimination and diversity in Hybrid Distill. Experiments on various downstream tasks, including classification, detection, and segmentation, also showcase its superiority.

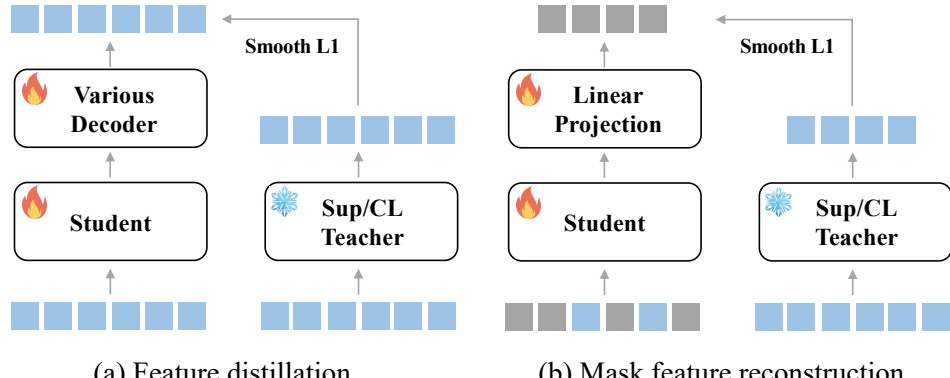

(a) Feature distillation          (b) Mask feature reconstruction

Figure 2: The distillation configurations inspected in Sec. 2. We examine the influence of (a) different decoders for feature distillation (no decoder, linear projection, asymmetric decoder), as well as replacing the feature distillation objective with (b) mask feature reconstruction. Blue and grey squares represent visible and mask tokens, respectively.

## 2   Model Evaluation: Diversity and Discrimination

This section re-examines the findings of previous feature distillation or mask feature reconstruction works (Liu et al., 2022; Wei et al., 2022b) illustrated in Sec. 1, and highlights their limitations in incorporating diversity and discrimination. **The distillation settings follows Liu et al. (2022); Wei et al. (2022b), where only features in the final layer of a ViT-B teacher (Dosovitskiy et al., 2020) are utilized as distillation objective. Smooth L1 loss is used during the distillation process.**. Different configurations we inspected are shown in Fig. 2.

### 2.1   Preliminary

**Definition of discrimination and diversity.** We first introduce the definitions of diversity and discrimination used to judge the representations.

● *Discrimination* means that the representations contain more global patterns tailored to object shapes, which is beneficial for recognizing objects and distinguishing images. A similar definition of discrimination can be found in Huang et al. (2022).

● *Diversity* means that the model pays attention to both local and global information and can acquire representations through more diverse attention with different attention distances, particularly in the last few layers. This notion of diversity is similarly defined and discussed in Wei et al. (2022b); Xie et al. (2022a); Park et al. (2023).

**Evaluation strategies.** We measure discrimination and diversity by per-head average attention distance (Dosovitskiy et al., 2020) and normalized mutual information (NMI) (Strehl & Ghosh, 2002). We also include additional linear probing evaluation in Sec. B.1 of the appendix.

● *The average attention distance,* which is also used in Liu et al. (2022); Wei et al. (2022b); Xie et al. (2022a); Park et al. (2023), calculates the distance between the query and the key tokens based on the attention weights and averages them by head in each transformer layer, providing insight of whether the attention is global or local. A low attention distance value means that queries attend to small regions near the query location, while a high value means queries have a larger receptive field.

● *The NMI metric,* utilized in Park et al. (2023), measures the attention is attending to different tokens or similar ones. A low NMI indicates attention maps are less reliant on the query tokens, implying that all queries focus on similar tokens. Let $p(q) = \frac{1}{N}$ represent the distribution of query tokens, where $N$ is the total token number. The joint distribution of query and key is computed as $p(q,k) = \pi(k|q)p(q)$, where $\pi(k|q)$ is the normalized self-attention matrix. Thus, NMI can be calculated by $\frac{I(q,k)}{\sqrt{H(q)H(k)}}$, where $I(q,k)$ is the mutual information and $H(\cdot)$ is the marginal entropy.

### 2.2   The Increase in Diversity Derives from the Asymmetric Designs

Fig. 3 (a)-(c) measures the average attention distance after feature distillation with various decoders appended the student encoder. Note that for the asymmetric decoder, we also visualize the average attention distance of the decoder for comparison (r.f. Fig. 3(c)), while the actual encoder layers are the first twelve layers. It can be seen that using no decoder or linear projection leads to a negligible increase (or even decrease) in attention diversity compared to teacher models (r.f. Fig. 1 (a)-(b)),

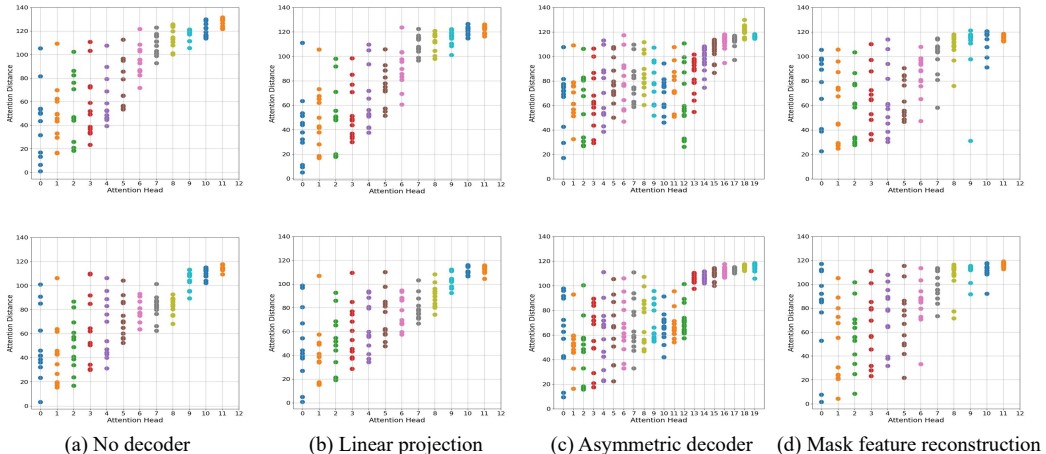

(a) No decoder   (b) Linear projection   (c) Asymmetric decoder   (d) Mask feature reconstruction

Figure 3: Average attention distance when using (a) no decoder, (b) linear projection, and (c) asymmetric decoder after the student encoder and setting (d) mask feature reconstruction as the learning objective. The first and the second rows are distilled using DeiT and CLIP teachers, respectively.

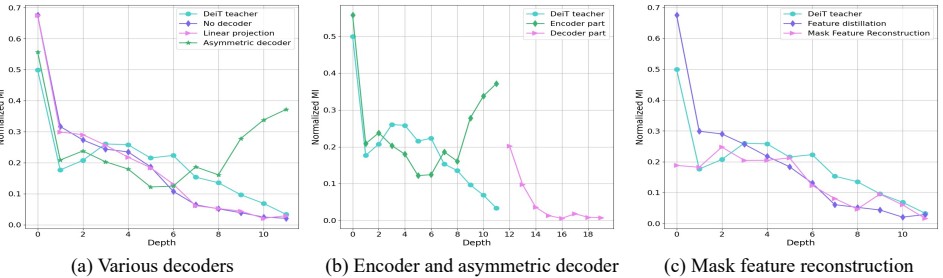

(a) Various decoders   (b) Encoder and asymmetric decoder   (c) Mask feature reconstruction

Figure 4: The normalized mutual information (NMI). We visualize (a) the encoder behavior when using various decoders and (c) the mask feature reconstruction objective. (b) further visualizes the behavior of the introduced asymmetric decoder based on (a).

reflecting that feature distilling itself cannot bring diversity. Adding extra attention layers at the decoder can make the student encoder more diverse, but hinders its discrimination since the last encoder layer (numbered 11) no longer captures long-range patterns. Fig. 4(a) further compares NMI using the DeiT teacher and the results are in line with Fig. 3, *i.e.*, without asymmetric designs, the student collapses into homogeneity and behaves similarly in the last few layers. Conversely, the use of asymmetric decoders greatly reduces discrimination of the encoder, as queries no longer pay attention to the main subjects in the last encoder layer.

Note that the above discussions focus on decoders, while for FD (Wei et al., 2022b), the asymmetric designs rise from adding additional learnable parameters and relative position bias at the encoder side. In Sec. B.2 of the appendix, we demonstrate that for FD, the increase in diversity also arises from these designs while the diversity brought by them is limited and not always significant.

## 2.3   THE ASYMMETRIC DECODER HARMS THE ENCODER DISCRIMINATION

Fig. 3(c) and Fig. 4(b) further measure the average attention distance and NMI of the asymmetric decoder. Our findings suggest that the decoder transfers the discrimination of the teacher, as its behavior is similar to that of the last few layers of the teacher model where queries pay attention to similar tokens. Reducing the number of decoder layers does not eliminate this transfer, as further demonstrated in Sec. B.3 of the appendix. Since only the student encoder is retained and applied to downstream tasks after distillation, the semantic information maintained is weakened, which explains why in dBOT (Liu et al., 2022), different teachers tend to yield similarly behaving models after multi-stage distillation. Note that dBOT conducts feature distilling in a mask reconstruction way, while we demonstrate in both Sec. 2.4 and the visualization in Sec. B.4 of the appendix that it behaves similarly to directly distilling features.

## 2.4   MASK RECONSTRUCTION OF HIGH-LEVEL SEMANTICS DOES NOT IMPROVE DIVERSITY

Fig. 3(d) and Fig. 4(c) examine the influence of mask reconstructing high-level information. To eliminate the effect of the asymmetric decoder, we feed both the masks and tokens into the encoder

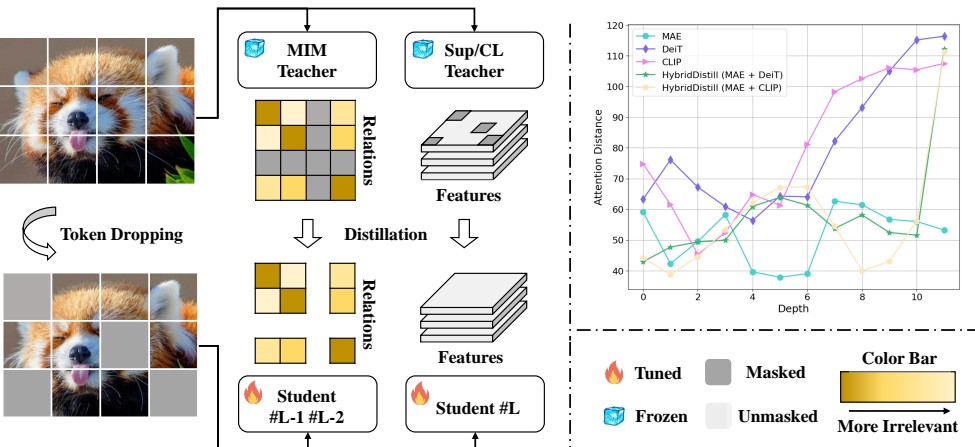

Figure 5: Hybrid Distill pipeline and its effectiveness in ensuring discrimination and diversity. Note that in this figure, we further average the per-head attention distance and obtain a single average attention distance for each layer to reflect the per-layer overall distance.

simultaneously and use only linear projection as the decoder. The overall process is thus similar to SimMIM (Xie et al., 2022b), except that we use the high-level information obtained from the supervised/CL teacher as the distilling objective. The attention distance and the NMI value prove that reconstructing high-level information brings no diversity gains towards directly distilling features, which is consistent with the finding of Xue et al. (2022), *i.e.*, reconstruction is unnecessary for MIM with semantic-rich teachers. This phenomenon also implies that the diversity of MIM mainly arises from the low-level reconstructing objective since diversity is absent in high-level reconstruction.

## 3 HYBRID DISTILLATION

From the above discussion, we conclude that existing single-teacher distillation pipelines have limitations in providing discrimination and diversity. In this section, we propose a hybrid distillation framework, termed Hybrid Distill, to ensure both discrimination and diversity.

### 3.1 OVERVIEW

Given a supervised/CL pre-trained model $T_c$, and a MIM pre-trained model $T_m$, Hybrid Distill simultaneously distills knowledge from these two different types of teachers, aims at combining their respective advantages to enhance the new representations in a randomly initialized student model $S_\theta$ where $\theta$ is its learnable parameters. ViT (Dosovitskiy et al., 2020) is adopted for all the models, and $T_m$ is provided by MAE (He et al., 2022) while $T_c$ is provided by DeiT (Touvron et al., 2021) or CLIP (Radford et al., 2021). Specifically, we employ a symmetric design in Hybrid Distill, *i.e.*, avoiding adding any types of decoder to the student model to maintain the discrimination. The overall Hybrid Distill framework is shown in Fig. 5 and its overall objective is:

$$\max_\theta \mathop{\mathbb{E}}_{x \sim \mathcal{X}} \alpha \mathcal{D} \left\{ T_c(x) \odot M, S_\theta(M \odot x) \right\}$$
$$+ \beta \mathcal{D} \left\{ T'_m(x) \odot M, S'_\theta(M \odot x) \right\}, \tag{1}$$

where $\odot$ is an element-wise product operation. $M$ is a mask provided by the teacher model using the strategy described in Sec. 3.2 and $M \odot x$ denotes the unmasked patches. $\mathcal{D}(\cdot, \cdot)$ is the distance measurement, and we use smooth L1 distance in our experiment. $\alpha$ and $\beta$ is the hyperparameter that controls the contribution of the two teachers. Note that we do not distill the final output features $T_m(x)$ from the MIM pre-trained model but instead use the token relations preceding the last layer, denoted as $T'_m(x)$, as the learning objective. $S'_\theta$ is similar to $T'_m(x)$, which represents the token relations of the student model. Details are illustrated in Sec. 3.2.

### 3.2 DISTILLING STRATEGIES

**What to distill?** Different from previous works (Wei et al., 2022b; Fang et al., 2022; Xue et al., 2022) that directly distill the features of teacher models, we analyze that the diversity of MIM arises from their superior token-level relationship modeling, while features from supervised/CL teachers excel at image-level discrimination. Hence, we apply different distilling targets to $T_c$ and $T_m$.

Figure 6: The (a) average attention distance, (b) NMI, and (c) attention visualization of the student model obtained from Hybrid Distill with MAE and CLIP teachers.

Specifically, taking $T_m$ as an example, we decompose $T_m$ into $T_m^1 \circ T_m^2 \circ \cdots \circ T_m^L$, where $T_m^i$ is the $i^{th}$ layer of $T_m$ and is composed of a multi-head self-attention (MSA) layer and an MLP layer. Given $x_m^i$ as the input of the $i^{th}$ layer, the calculation in $T_m^i$ can be represented as:

$$\begin{aligned} \mathrm{R}_m^i(x_m^i) &= Q_m^i(x_m^i) K_m^i(x_m^i)^T, \\ \mathrm{MSA}_m^i(x_m^i) &= \mathrm{Softmax}\left(\mathrm{R}_m^i(x_m^i)/\sqrt{d}\right) V_m^i(x_m^i), \\ T_m^i(x_m^i) &= x_m^i + \mathrm{MLP}(x_m^i + \mathrm{MSA}_m^i(x_m^i)), \end{aligned} \tag{2}$$

where $Q_m^i$, $K_m^i$, and $V_m^i$ denotes the linear mappings for $x_m^i$ and $d$ equals to the dimension of $x_m^i$. Then, for MIM teacher $T_m$, we set the token relation $\mathrm{R}_m^i(x_m^i)$ as the distilling target, while for supervised/CL teacher $T_c$, we set the output features $T_c^i(x_c^i)$ as the target.

**Where to distill?** As shown in Fig. 1(a)-(b), supervised and CL models tend to collapse into homogeneity in the last few layers, so Hybrid Distill chooses to distill token relations from $T_m$ in these layers to address this collapse and improve diversity. While for the last layer of $S$ which is crucial for discrimination, Hybrid Distill directly distills knowledge from $T_c$ using the output features. Specifically, we distill token relations from $T_m$ at the $L-1$ and $L-2$ layers and distill features from $T_c$ at the $L$ layer of ViT. Accordingly, the learning objective $T_c(x)$ and $T_m'(x)$ in Eq. 1 become:

$$\begin{aligned} T_c(x) &= T_c^L(x), \\ T_m'(x) &= [R_m^{L-1}(x), R_m^{L-2}(x)]. \end{aligned} \tag{3}$$

**Distillation acceleration.** Since tokens are redundant in an image, it is promising to mask some unnecessary tokens of the student model $S$ to reduce memory and time costs. We also find that removing redundant tokens can play a regulatory role, helping the model avoid local optima during the distillation process. Specifically, we use the MIM teacher $T_m$ to guide the identification of redundant tokens and provide the token mask. Inspired by Li et al. (2023), we propose a progressive redundant token masking strategy, which generates token masks at different layers of $T_m$ in a progressive manner. Given token sequence $x_m^i$ and the mask $M_m^{i-1}$ provided by the previous layer, we define the tokens in $x_m^i \odot M_m^{i-1}$ and are top $K\%$ similar to their average token as redundant tokens in the $i^{th}$ layer and generate a redundant token mask for them. The above process is denoted as $T(x_m^i \odot M_m^{i-1}, K)$. Next, we update $M_m^i$ using $T(x_m^i \odot M_m^{i-1}, K)$ and $M_m^{i-1}$ as follows:

$$M_m^i = \begin{cases} M_m^{i-1} - T(x_m^i \odot M_m^{i-1}, K), & \text{if } i \in I, \\ M_m^{i-1} & \text{if } i \notin I. \end{cases} \tag{4}$$

where $I$ is the set of layers required to update the token mask. For $M_m^0$, all elements are set to 1. Finally, we set the mask $M$ for the student model as $M = M_m^L$.

### 3.3 PROPERTY ANALYSIS

**Average attention distance.** Fig. 6(a) visualizes the average attention distance of the student model with CLIP and MAE as teachers, while the visualization of CLIP and MAE teachers are shown in Fig. 1. These visualizations demonstrate that Hybrid Distill enhances the discrimination ability of the student model, compensating for the semantic lacking of the MAE teacher. Moreover, Hybrid Distill avoids succeeding attention collapse from the CLIP teacher and generates more diverse representations in the last few layers.

**Normalized mutual information.** Fig. 6(b) further inspects the NMI. The results demonstrate that the mutual information between tokens is significantly enhanced in the layers where the MAE

Table 1: Main results on ImageNet-1K classification, COCO detection and instance segmentation, and ADE20K semantic segmentation. $\star$: using MAE+DeiT teachers. $\dagger$: using pretrained MAE+CLIP teachers. $\ddagger$: using ImageNet-21k dataset for distillation.

| Method | Backbone | Distill. | IN-1K | COCO | | ADE20K |
| | | | | $AP^{box}$ | $AP^{Mask}$ | |
|---|---|---|---|---|---|---|
| DeiT (Touvron et al., 2021) | ViT-B | | 81.8 | 46.9 | 41.5 | 47.0 |
| MoCo v3 (Chen et al., 2021) | | | 83.2 | 45.5 | 40.5 | 47.1 |
| DINO (Caron et al., 2021) | | | 83.3 | 46.8 | 41.5 | 47.2 |
| MAE (He et al., 2022) | | | 83.6 | 48.4 | 42.6 | 48.1 |
| CAE (Chen et al., 2022a) | | | 83.3 | 48.0 | 42.3 | 47.7 |
| SdAE (Chen et al., 2022b) | | | 84.1 | 48.9 | 43.0 | 48.6 |
| CLIP (Radford et al., 2021) | | | 83.6 | 47.6 | 42.3 | 49.6 |
| Distill-DeiT | ViT-B | $\checkmark$ | 82.0 | 47.7 | 42.1 | 47.3 |
| Distill-MAE | | | 83.7 | 49.1 | 43.1 | 47.8 |
| Hybrid Distill$\star$ | | | **83.7** | **50.3** | **44.2** | **49.1** |
| Distill-CLIP | ViT-B | $\checkmark$ | 84.8 | 49.5 | 43.5 | 50.3 |
| Hybrid Distill$\dagger$ | | | **85.1** | **50.6** | **44.4** | **51.5** |
| MAE (He et al., 2022) | ViT-L | | 85.9 | 54.0 | 47.1 | 53.6 |
| CLIP (Radford et al., 2021) | | | 86.1 | 52.7 | 46.2 | 54.2 |
| Hybrid Distill$\dagger$ | ViT-L | $\checkmark$ | **87.6** | **54.4** | **47.4** | **55.9** |
| Hybrid Distill$\dagger\ddagger$ | ViT-L | $\checkmark$ | **88.0** | **54.6** | **47.6** | **56.3** |

token relationships are distilled. Besides, this enhancement does not compromise the discrimination obtained from CLIP, as evidenced by attention in the final layers still attending to similar tokens.

**Attention visualization.** Fig. 6(c) further visualizes the attention between a given query and other keys at different layers to examine behaviors. Compared to MAE, Hybrid Distill exhibits better discrimination ability, *i.e.*, the query tokens of the last layer have global attention towards the main object of the images, regardless of their location. Besides, Hybrid Distill also improves the locality of the model in the $10^{th}$ layer, where attention collapse is known to occur in the CLIP teacher.

## 4 EXPERIMENTS

### 4.1 IMPLEMENTATION DETAILS

Our experiments are conducted on 8 V100 GPUs. The batch size, learning rate, and weight decay are set to 1024, 6e-4, and 0.05, respectively. AdamW (Loshchilov & Hutter, 2017) optimizer and cosine decay (Loshchilov & Hutter, 2016) schedule is used. The input size is $224^2$. For ViT-B, the distillation is based on ImageNet-1K Russakovsky et al. (2015), and the epoch is 300 for main results and 100 for ablation studies. For ViT-L, we conduct 300 epoch distillation based on ImageNet-1K and 40 epoch distillation based on ImageNet-21K, respectively. The hyperparameter $\alpha$ and $\beta$ are set to 1.0 and the redundant token masking set $I$ is set to $[0, L/3, 2L/3]$ following Li et al. (2023). The performances are tested on different downstream tasks, including ImageNet-1K, CIFAR100 (Krizhevsky et al., 2009), Cars (Krause et al., 2013), and iNaturalist19 (Van Horn et al., 2018) classification, COCO (Lin et al., 2014) object detection and instance segmentation, and ADE20K (Zhou et al., 2019) segmentation. More downstream details are included in the appendix.

### 4.2 MAIN RESULTS

This section presents benchmark results of Hybrid Distill on different downstream. We also list results for supervised and self-supervised pre-trained models, as well as 300-epoch uni-distillation baselines which use the same symmetrical structures as Hybrid Distill for comparison. As shown in Tab. 1, **Hybrid Distill achieves performance gains on all downstream tasks, especially for the dense-level ones that rely more on diversity.** Specifically, although the performance of DeiT is suboptimal, its strength can be complementary to MAE and brings considerable benefits, *i.e.*, when using DeiT and MAE teachers, Hybrid Distill achieves 50.3 $AP^{box}$ and 44.2 $AP^{mask}$ on COCO, as well as 49.1 mIoU on ADE20K, surpassing Distill-MAE by 1.2, 1.1, and 1.3, respectively. Similarly, Hybrid Distill achieves 50.6 $AP^{box}$ and 44.4 $AP^{mask}$ on COCO, as well as 51.5 mIoU on ADE20K when using CLIP and MAE teachers, outperforming Distill-CLIP by 1.1, 0.9, and 1.2, respectively. When using the ViT-L backbone and larger-scale ImageNet-21K dataset for distillation,

Table 2: Classification results on CIFAR100, Cars and INautralist19. $\star$: using MAE+DeiT teachers. $\dagger$: using MAE+CLIP teachers. $\ddagger$: using ImageNet-21k dataset for distillation.

| Method | Backbone | CIFAR100 | Cars | INaturalist19 | Mean |
|---|---|---|---|---|---|
| DeiT (Touvron et al., 2021) | ViT-B | 91.4 | 92.0 | 77.3 | 86.9 |
| MAE (He et al., 2022) | ViT-B | 89.6 | 89.5 | 75.2 | 84.8 |
| Distill-DeiT | ViT-B | 91.2 | 92.5 | 78.3 | 87.3 |
| Distill-MAE | ViT-B | 90.3 | 93.1 | 79.0 | 87.5 |
| Hybrid Distill$^\star$ | ViT-B | **91.7** | **94.1** | **80.2** | **88.7** |
| Distill-CLIP | ViT-B | 91.6 | 94.3 | 81.6 | 89.2 |
| Hybrid Distill$^\dagger$ | ViT-B | **92.0** | **94.5** | **81.9** | **89.5** |
| Hybrid Distill$^{\dagger\ddagger}$ | ViT-L | **94.5** | **95.6** | **85.3** | **91.8** |

Table 3: Different combinations of two teacher models. $T_c(x)$: DeiT, $T_m(x)$: MAE.

| Targets | $AP^{box}$ | $AP^{mask}$ |
|---|---|---|
| $T_c(x)$ | 47.5 | 41.8 |
| $T_m(x)$ | 48.9 | 43.1 |
| $T_c(x) + T_c'(x)$ | 46.8 | 41.5 |
| $T_m(x) + T_m'(x)$ | 48.9 | 43.2 |
| $T_c(x) + T_m'(x)$ | **50.0** | **43.9** |

Table 4: Different combinations of two teacher models. $T_c(x)$: CLIP, $T_m(x)$: MAE. $\star$: using the ImageNet-100 pretrained weights.

| Targets | $AP^{box}$ | $AP^{mask}$ |
|---|---|---|
| $T_c(x)$ | 49.1 | 43.1 |
| $T_m(x)$ | 48.9 | 43.1 |
| $T_c(x) + T_c'(x)$ | 49.1 | 43.2 |
| $T_c(x) + T_m'(x)$ | **50.4** | **44.1** |
| $T_c(x) + T_m'(x)^\star$ | 49.5 | 43.5 |

the performance can be further boosted to 54.6 $AP^{box}$, 47.6 $AP^{mask}$ and 56.3 mIoU on respective tasks. Hybrid Distill does not bring noticeable gains on the ImageNet-1K dataset, which is in line with our expectations since diversity is of limited help for image-level classification, especially when the distillation and downstream data distribution are the same (Xie et al., 2022a). However, the still comparable results towards Distill-CLIP reflect that Hybrid Distill increases diversity without sacrificing discrimination. Besides, we prove in Tab. 2 that diversity can also bring benefits to image classification when distillation and downstream data distribution are different, *i.e.*, Hybrid Distill achieves more significant gains on several small-scale classification datasets.

## 4.3 Ablation Study

**Different combinations of two teachers.** We first evaluate the benefits of combining two teachers for distillation. As shown in Tab. 3, adding additional MAE attention regularization can bring noticeable improvements (2.5 on $AP^{box}$ and 2.1 on $AP^{mask}$) compared to directly distilling from the DeiT teacher. Moreover, the additional attention regularization cannot bring benefits when only using a single DeiT teacher, which suggests that the benefits come from the introduction of MAE teacher. The above conclusions are consistent when using CLIP and MAE teachers as illustrated in Tab. 4. We also try a much weaker version of MAE teacher which is only pre-trained on ImageNet-100 for 100 epochs in Tab. 4. We lower the weight of this teacher to avoid its impact on discrimination. The results are still positive, which reflects the power of the MIM pre-training in modeling diversity.

**Distilling target of the MIM teacher.** We then examine the distilling target of the MIM teacher. As shown in Tab. 5, distilling the relation $R_m^i$ brings the best detection performance ($50.0 AP^{box}$). Distilling $MSA_m^i$ achieves a close performance ($49.8 AP^{box}$) since its essential is also distilling relationships, while directly distilling the feature maps $T_m^i$ brings the worst performance ($49.6 AP^{box}$). Nevertheless, all these schemes outperform the DeiT distillation baseline, and the trends are consistent when using CLIP and MAE teachers, as shown in Tab. 6. Besides, we also evaluate a basic setting that directly distills the features of both the MIM and supervised/CL teachers at the last layer. The results are not satisfactory especially when the weaker DeiT teacher is used. The above results highlight the effectiveness of the designs in Hybrid Distill.

**Distilling position of the MIM teacher.** Tab. 7 inspect the distilling position of the MIM teacher. We first experiment with distilling MAE relations at the front, middle, and back layers. Distilling at the back layers achieves better results, *i.e.*, $1.5 AP^{box}$ and $2.4 AP^{box}$ gains towards distilling at the front and middle, respectively. The results are consistent with the fact that attention collapse tends to occur in these back layers. We then ablate the number of distilling layers and find that distilling at the two layers preceding the final layer (*i.e.*, 10,11) contributes to the best results.

**Token masking strategy.** Tab. 8 studies different masking strategies for the student model. Since we progressive drop the redundant tokens three times, the actual tokens used in the student model

Table 5: The distilling targets of $T'_m(x)$. $T_c(x)$: DeiT, $T_m(x)$: MAE. $\star$ means distilling MAE and DeiT features at the last layer.

| Targets | $AP^{box}$ | $AP^{mask}$ |
|---|---|---|
| $T_m^{i}$ $\star$ | 47.7 | 42.1 |
| $T_m^{i}$ | 49.6 | 43.5 |
| $MSA_m^{i}$ | 49.8 | 43.7 |
| $R_m^{i}$ | **50.0** | **43.9** |

Table 6: The distilling targets of $T'_m(x)$. $T_c(x)$: CLIP, $T_m(x)$: MAE.$\star$ means distilling MAE and CLIP features at the last layer.

| Targets | $AP^{box}$ | $AP^{mask}$ |
|---|---|---|
| $T_m^{i}$ $\star$ | 49.7 | 43.7 |
| $T_m^{i}$ | 49.9 | 44.0 |
| $MSA_m^{i}$ | 50.1 | 44.0 |
| $R_m^{i}$ | **50.4** | **44.1** |

Table 7: The distilling position of $T_m$.

| Distilling layers | $AP^{box}$ | $AP^{mask}$ |
|---|---|---|
| 1-11 | 48.8 | 43.0 |
| 1,2,3 | 47.4 | 41.9 |
| 5,6,7 | 48.3 | 42.7 |
| 9,10,11 | 49.8 | 43.7 |
| 10,11 | **50.0** | **43.9** |
| 11 | 49.2 | 43.3 |

Table 8: The token masking strategy.

| Strategy | Ratio | $AP^{box}$ | $AP^{mask}$ |
|---|---|---|---|
| No | 100% | **50.0** | **43.9** |
| Random | 35% | 49.2 | 43.3 |
| Direct | 35% | 49.6 | 43.7 |
| Progressive | $13\%(50\%^3)$ | 48.4 | 42.8 |
| Progressive | $34\%(70\%^3)$ | 49.9 | 43.8 |
| Progressive | $73\%(90\%^3)$ | 49.9 | 43.8 |

are $(1-K)^3\%$. We observe that when dropping 30% tokens at a time, Hybrid Distill achieves very close performance ($49.9AP^{box}$ and $43.8AP^{mask}$) to the no masking results and outperforms the random masking strategy and the direct masking strategy which only generates token mask at the last layer. In addition, we notice that our token masking strategy also has a regularizing effect, which can prevent the model from falling into a locally optimal when training for longer epochs. Details about this effect are included in the appendix.

## 5 RELATED WORK

**Model pre-training.** Contrastive learning (CL) (Chen et al., 2020a; He et al., 2020; Chen et al., 2020b; Grill et al., 2020) and masked image modeling (MIM) (Bao et al., 2022; Xie et al., 2022b; He et al., 2022) dominate the recent pre-training research. The former is achieved by pulling close the features of two different augment views. While the latter, inspired by masked language modeling (Kenton & Toutanova, 2019; Zhang et al., 2019), is realized by reconstructing the mask part of the input. Recently, multi-model extensions (Radford et al., 2021; Cui et al., 2022; Li et al., 2022) of the CL pre-training have also been proposed by utilizing the image-text pairs. These different types of pre-training frameworks are proven to have different properties (Park et al., 2023; Xie et al., 2022a), and Huang et al. (2022); Zhou et al. (2022); Jiang et al. (2023) further try to use both reconstruction loss and contrastive loss to combine their respective advantages. Different from them, we resort to distillation and exploit off-the-shelf teachers to achieve better representations.

**Knowledge distillation.** Knowledge distillation (Park et al., 2019; Tian et al., 2019; Romero et al., 2014) utilizes a well-trained teacher to guide the feature learning of the student model, thus transferring its ability to the student. Some recent works (Wei et al., 2022b; Fang et al., 2022; Wang et al., 2021; Wei et al., 2022a; Peng et al., 2022) propose to utilize it to extend existing pretrained models or paradigms. Feature distillation (FD) (Wei et al., 2022b) finds that distilling the feature map of the supervised/CL pretrained teacher can bring diverse representation to the student and make it more friendly for downstream fine-tuning. dBOT (Liu et al., 2022), MVP (Wei et al., 2022a), and BEiT v2 (Peng et al., 2022) change the mask reconstruction object of MIM to the knowledge of the teacher model to boost MIM pre-training with semantic information. MILAN (Hou et al., 2022) introduces a prompting decoder design to improve the distillation paradigm. In this paper, we analyze their properties and propose a new hybrid distillation framework to deal with their deficiencies.

## 6 CONCLUSION

This paper proposed a hybrid distillation framework that simultaneously distills knowledge from the supervised/CL and MIM pre-trained teacher. The framework addresses the limitations of single-teacher distillation, where diversity and discrimination can not be ensured simultaneously. Specifically, Hybrid Distill carefully designs the distilling target and location, *i.e.*, distilling relations from MIM in layers where attention collapse tends to occur and distilling features from supervised/CL in the last layer to preserve discrimination. A progressive redundant token masking strategy is also proposed for reducing the distilling costs. Experiments prove that Hybrid Distill can acquire better properties and achieve promising results on various downstream.

**Acknowledgment** This work was supported in part by the National Natural Science Foundation of China under Grant 62125109, Grant 62250055, Grant 61931023, Grant 61932022, Grant 62371288, Grant 62320106003, Grant 62301299, Grant T2122024, Grant 62120106007.

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

## A MORE EXPERIMENTAL RESULTS

### A.1 COMPARED WITH MORE BASELINES

Tab. 9 compares Hybrid Distill with two other methods, *i.e.,* dBOT (Liu et al., 2022) and FD (Wei et al., 2022b), which employ asymmetric designs in distillation. We conduct distilling for 300 epochs

Table 9: Compared with more baselines using ViT-B as the backbone. ⋆: using MAE+DeiT teachers. †: using MAE+CLIP teachers.

| Method | COCO | | ADE20K |
| --- | --- | --- | --- |
| | $AP^{box}$ | $AP^{Mask}$ | |
| Distill-DeiT | 47.7 | 42.1 | 47.3 |
| Distill-MAE | 49.1 | 43.1 | 47.8 |
| Distill-CLIP | 49.5 | 43.5 | 50.3 |
| FD-DeiT (Wei et al., 2022b) | 47.0 | 41.6 | 47.9 |
| FD-MAE (Wei et al., 2022b) | 48.1 | 42.6 | 47.0 |
| FD-CLIP (Wei et al., 2022b) | 49.2 | 43.3 | 50.5 |
| dBOT-DeiT (Liu et al., 2022) | 47.5 | 41.9 | 47.9 |
| dBOT-MAE (Liu et al., 2022) | 49.3 | 43.5 | 48.2 |
| Hybrid Distill⋆ | 50.3 | 44.2 | 49.1 |
| Hybrid Distill† | **50.6** | **44.4** | **51.5** |

Table 10: Object detection and instance segmentation results with *Cascade Mask-RCNN*. ⋆: using MAE+DeiT teachers. †: using MAE+CLIP teachers.

| Method | Epoch | $AP^{box}$ | $AP^{mask}$ |
| --- | --- | --- | --- |
| Distill-DeiT | 300 | 50.4 | 43.4 |
| Distill-MAE | 300 | 51.9 | 44.7 |
| Distill-CLIP | 300 | 52.4 | 45.0 |
| dBOT-DeiT (Liu et al., 2022) | $2 \times 800$ | 52.5 | - |
| dBOT-MAE (Liu et al., 2022) | $2 \times 800$ | 52.7 | - |
| dBOT-CLIP (Liu et al., 2022) | $1 \times 1600$ | **53.6** | - |
| Hybrid Distill⋆ | 300 | 53.0 | 45.6 |
| Hybrid Distill† | 300 | 53.4 | **45.9** |

based on their corresponding official codes[1]. We omit the dBOT-CLIP result since dBOT specifically removes the asymmetric designs for CLIP, thus its distillation process is similar to our Distill-CLIP baseline. As shown in Tab. 9, their benefits towards symmetrical distillation (Distill-X) are not always significant, and the performance is inferior to our Hybrid Distill, which validates the effectiveness of our framework.

## A.2 RESULTS WITH CASCADE MASK-RCNN

Tab. 10 further presents the object detection and instance segmentation results of Hybrid Distill with Cascade Mask-RCNN, which allows for a direct comparison with dBOT (Liu et al., 2022), as they also provide 1600-epoch distillation results under this setting. As shown, 300-epoch Hybrid Distill with MAE and DeiT teachers can achieve 53.0 $AP^{box}$, outperforming 1600-epoch dBOT-DeiT (52.5 $AP^{box}$) and dBOT-MAE (52.7 $AP^{box}$). Additionally, 300-epoch Hybrid Distill with MAE and CLIP teachers achieves 53.4 $AP^{box}$, which is also very close to the 1600-epoch dBOT-CLIP result (53.6 $AP^{box}$). The above results reflect that due to the better properties obtained, Hybrid Distill can obtain promising results with fewer distilling epochs.

## A.3 LINEAR PROBING RESULTS

Tab. 11 shows the linear probing results on ImageNet-1K. The subpar performance of dBOT further validates our perspective discussed in Sec. 2.3, which suggests that its increase in diversity comes at the expense of sacrificing discrimination. In contrast, Hybrid Distill consistently maintains outstanding linear probing performance, thus confirming the effectiveness of our approach in simultaneously promoting diversity and discrimination.

---

[1]dBOT (Liu et al., 2022): https://github.com/liuxingbin/dbot/. FD (Wei et al., 2022b): https://github.com/SwinTransformer/Feature-Distillation/. Since FD does not provide codes for downstream verification, we uniformly perform verification under our downstream frameworks.

Table 11: Results for linear probing on ImageNet-1K. $\star$: using MAE+DeiT teachers. $\dagger$: using MAE+CLIP teachers.

| Method | Linear Probing |
|---|---|
| MAE | 67.8 |
| dBOT | 67.9 |
| Hybrid Distill$^\star$ | **80.9** |
| Hybrid Distill$^\dagger$ | 80.1 |

Table 12: Hybrid Distill uses MAE and DINO as teachers. Object Detection and instance segmentation results are reported with *Mask-RCNN*, following the setting in Tab. 1 of our main paper.

| Method | $AP^{box}$ | $AP^{mask}$ |
|---|---|---|
| MAE (He et al., 2022) | 48.4 | 42.6 |
| DINO (Caron et al., 2021) | 46.8 | 41.5 |
| Distill-DINO | 47.5 | 41.9 |
| Distill-MAE | 49.1 | 43.1 |
| Hybrid Distill | **49.6** | **43.5** |

### A.4 HYBRID DISTILLATION WITH DINO

Tab. 12 test the results of our Hybrid Distill using the MAE (He et al., 2022) and DINO (Caron et al., 2021) teachers. Under this setting, Hybrid Distill achieves 49.6 $AP^{box}$ and 43.5 $AP^{mask}$. Although still superior to the baselines, results with DINO are not as good as those with CLIP and DeiT. We analyze that this is because the discrimination of DINO is weaker than DeiT and CLIP, which makes its complementarity with MAE also weaker than the latter two. The visualization in Fig.7 provides evidence for this. On the one hand, we notice that the average attention distance of DINO itself is lower than that of DeiT and CLIP in the final layer. On the other, the attention maintenance of the final layer after distillation is weaker compared with that obtained by DeiT and CLIP.

### A.5 MORE ABLATION STUDIES

**The choice of hyperparmeter $\alpha$ and $\beta$.** Tab. 13 and Tab. 14 ablate different setting of $\alpha$ and $\beta$, respectively . It can be concluded that combining two teachers for hybrid distillation can lead to performance gains towards only using one teacher, regardless of the value of hyperparameters $\alpha$ and $\beta$. Besides, we note that ensuring the weights of the more powerful model among the two teacher models ( *e.g.*, MAE for MAE+DeiT teachers and CLIP for MAE+CLIP teachers) count for better performance, and setting both $\alpha$ and $\beta$ to 1 for simplicity can already yield satisfactory results.

**Token masking strategy and local optima.** Tab. 15 further reveals that the proposed progressive redundant token masking strategy in Hybrid Distill can prevent the student from falling into local optima. As shown, when the token mask is removed and the distillation epoch is prolonged from 100 to 300, no further performance gains are observed. A similar phenomenon has also been observed in Fang et al. (2022). We analyze that over-fitting is the root cause of this problem and introducing token masks can alleviate it since they can play a regulatory role. The performance gains achieved by the token masks provide clear support for their effectiveness.

## B FURTHER DISCUSSION ABOUT DIVERSITY AND DISCRIMINATION

### B.1 LINEAR PROBING EVALUATION FOR JUDGING THE DISCRIMINATION

This section further supplements the linear probing results to judge the discrimination of the distillation configurations mentioned in Sec. 2. The results in Tab. 16 are consistent with the analysis presented in Sec. 2, indicating that the asymmetric decoder negatively impacts the discrimination (Sec. 2.3) and the behavior of mask feature reconstruction is similar to that of directly distilling features (Sec. 2.4). These results further corroborate our analysis of discrimination.

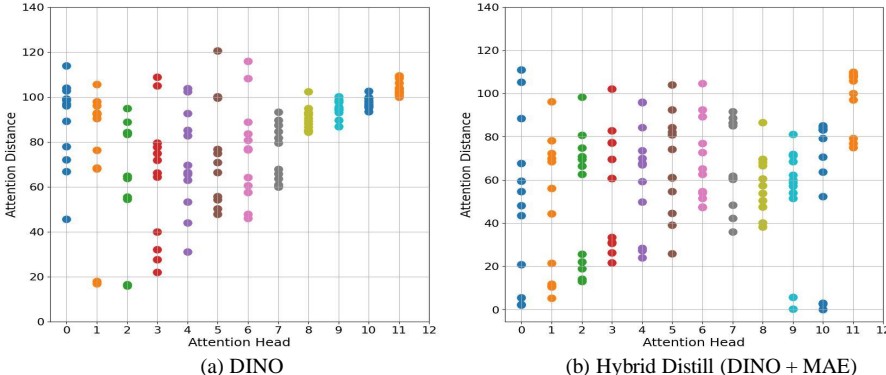

(a) DINO          (b) Hybrid Distill (DINO + MAE)

Figure 7: Average attention distance of different (a) DINO baseline and (b) Hybrid Distill with MAE and DINO as teachers.

Table 13: Ablation on the hyperparameter $\alpha$ (with $\beta = 1$) which controls the contribution of the supervised/CL teacher.

(a) $T_c(x)$: DeiT, $T_m(x)$: MAE.

| $\alpha$ | 0 | 0.1 | 0.3 | 0.5 | 0.7 | 1.0 |
|---|---|---|---|---|---|---|
| $\mathrm{AP^{box}}$ | 48.9 | 49.2 | 49.7 | 49.9 | **50.0** | **50.0** |
| $\mathrm{AP^{mask}}$ | 43.1 | 43.5 | 43.8 | **43.9** | **43.9** | **43.9** |

(b) $T_c(x)$: CLIP, $T_m(x)$: MAE.

| $\alpha$ | 0 | 0.1 | 0.3 | 0.5 | 0.7 | 1.0 |
|---|---|---|---|---|---|---|
| $\mathrm{AP^{box}}$ | 48.9 | 49.3 | 49.6 | 49.9 | 50.1 | **50.4** |
| $\mathrm{AP^{mask}}$ | 43.1 | 43.5 | 43.7 | 43.9 | 43.9 | **44.1** |

## B.2 ASYMMETRIC ENCODER DESIGNS

Fig. 8 studies the asymmetric encoder designs used in FD (Wei et al., 2022b), *i.e.*, adding additional learnable parameters and relative position bias to the attention layers of the student. As shown, the asymmetric encoder (Fig. 8(c)) can improve diversity compared to using only the symmetric encoder (Fig. 8(b)). However, compared to the DeiT teacher (Fig. 8(a)), it does not bring noticeable diversity gains. A similar visualization can be found in Wei et al. (2022b) (Figure 2 in their paper), and its diversity improvement is also not notable when using DeiT and CLIP teachers. Therefore, we conclude that the diversity brought by the asymmetric encoder is not always significant.

## B.3 REDUCING THE NUMBER OF THE ASYMMETRIC DECODER LAYERS

Fig. 9 investigates the effect of reducing the number of asymmetric decoder layers. We find that even with a reduced number of decoder layers, the discrimination in the last layer of the encoder still cannot be maintained. Therefore, we abandon this asymmetric decoder design in our Hybrid Distill to avoid losing discrimination.

## B.4 MASK FEATURE RECONSTRUCTION WITH ASYMMETRIC DECODER

Fig. 10 compares two variants of dBOT, *i.e.*, with the same asymmetric decoder design in dBOT but conducting direct feature distillation and mask feature reconstruction, respectively. It can be seen that the two tasks bring no significant differences, *i.e.*, the diversity is increased and the discrimination is lost regardless of the task. These visualizations further support our claim in Sec. 2.3 and Sec. 2.4 of our main paper.

Table 14: Ablation on the hyperparameter $\beta$ (with $\alpha = 1$) which controls the contribution of the MIM teacher.

(a) $T_c(x)$: DeiT, $T_m(x)$: MAE.

| $\beta$ | 0 | 0.1 | 0.3 | 0.5 | 0.7 | 1.0 |
|---|---|---|---|---|---|---|
| AP$^{\text{box}}$ | 47.5 | 48.2 | 49.3 | 49.3 | 49.5 | **50.0** |
| AP$^{\text{mask}}$ | 41.8 | 42.6 | 43.4 | 43.4 | 43.5 | **43.9** |

(b) $T_c(x)$: CLIP, $T_m(x)$: MAE.

| $\beta$ | 0 | 0.1 | 0.3 | 0.5 | 0.7 | 1.0 |
|---|---|---|---|---|---|---|
| AP$^{\text{box}}$ | 49.1 | 49.9 | 49.8 | 50.1 | 50.2 | **50.4** |
| AP$^{\text{mask}}$ | 43.1 | 43.8 | 43.8 | 43.9 | 44.1 | **44.1** |

Table 15: The token masking strategy for alleviating over-fitting. $\star$: using MAE+DeiT teachers. $\dagger$: using MAE+CLIP teachers.

| Method | Epoch | Masking | AP$^{\text{box}}$ | AP$^{\text{mask}}$ |
|---|---|---|---|---|
| Hybrid Distill$^{\star}$ | 100/300 | | **50.0**/50.0 | 43.9/**44.0** |
| Hybrid Distill$^{\star}$ | 100/300 | ✓ | 49.9/**50.3** | 43.8/**44.2** |
| Hybrid Distill$^{\dagger}$ | 100/300 | | **50.4**/50.3 | **44.1/44.1** |
| Hybrid Distill$^{\dagger}$ | 100/300 | ✓ | 50.2/**50.6** | 43.9/**44.4** |

Table 16: Results for linear probing on ImageNet-1K when using DeiT teachers.

| Distillation configuration | Linear probing |
|---|---|
| No decoder (Fig. 3 (a)) | 81.4 |
| Linear projection (Fig. 3 (b)) | 81.7 |
| Asymmetric decoder(Fig. 3 (c)) | 66.2 |
| Mask feature reconstruction (Fig. 3 (d)) | 79.5 |

## C  Implementation Details for Different Downstream Tasks

**Classification.** We report the fine-tuning results on ImageNet-1K. Following dBOT (Liu et al., 2022), the learning rate is set to 3e-4 and the batch size is set to 256. We also report results on CIFAR100 (Krizhevsky et al., 2009), Cars (Krause et al., 2013), and iNaturalist19 (Van Horn et al., 2018). For these datasets, the batch size is 768 and the learning rate is 7.5e-6.

**Object detection and instance segmentation.** Following Chen et al. (2022a), we fine-tune the student model on COCO (Lin et al., 2014) using the Mask-RCNN (He et al., 2017) framework. We train the network with the 1x schedule and the learning rate is set to 3e-4 for ViT-B and 2e-4 for ViT-L. We also provide the 1x results using the Cascade Mask-RCNN framework in the appendix, and the learning rate is set to 3e-4.

**Semantic segmentation.** The semantic segmentation evaluation is conducted on ADE20K (Zhou et al., 2019). Following Chen et al. (2022a;b), we use ViT (Dosovitskiy et al., 2020) with UperNet (Xiao et al., 2018) framework and fine-tune the model for 160k iterations. The batch size, learning rate, and weight decay are set to 16, 4e-4, and 0.05, respectively.

## D  Discussion with Other Distillation Methods

Compared to previous distillation methods (Wei et al., 2022b; Fang et al., 2022; Liu et al., 2022; Wei et al., 2022a; Peng et al., 2022), Hybrid Distill stands out by not being restricted to using a single teacher network. In addition to addressing the limitations of single-teacher distillation in enriching both diversity and discrimination (as discussed in Sec. 2), a more direct factor is that single-teacher distillation cannot create new knowledge, *e.g.*, creating additional discrimination for

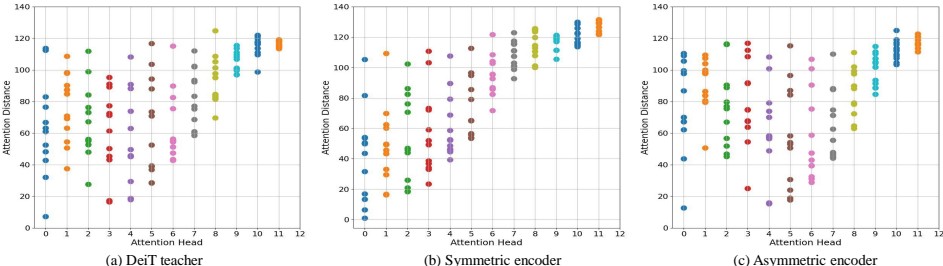

Figure 8: Average attention distance of (a) DeiT teacher and student models with (b) symmetric encoder and (c) asymmetric encoder.

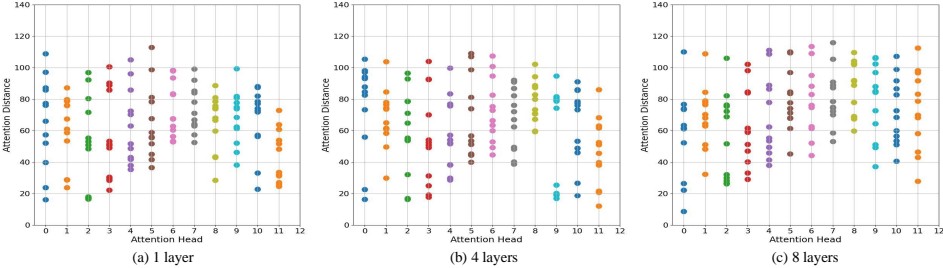

Figure 9: Average attention distance of using (a) 1, (b) 4, and (c) 8 asymmetric decoder layers, respectively.

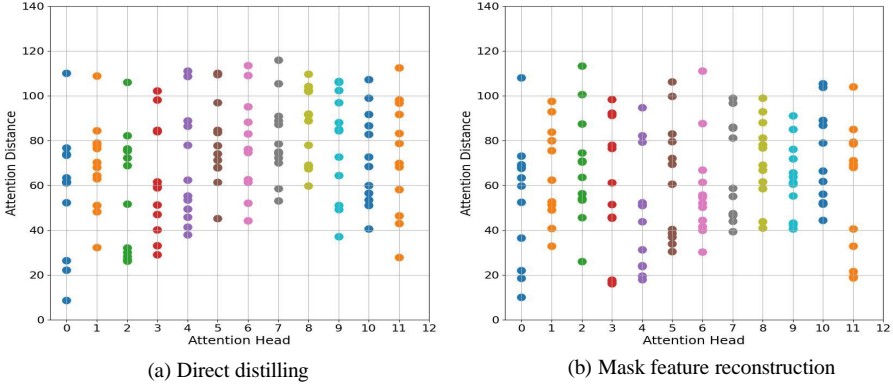

Figure 10: Average attention distance of different dBOT variants that conduct (a) direct feature distillation and (b) mask feature reconstruction, respectively.

Table 17: One-epoch training time and memory usage. Tested on 8 V100 with 128 per-GPU batch size.

| Method | One-epoch training time | GPU memory |
|---|---|---|
| MAE (He et al., 2022) | 9min15s | 17807M |
| dBOT (Liu et al., 2022) | 12min42s | 18953M |
| Hybrid Distill | 11min12s | 9503M |

the student model when using the MIM teacher. Therefore, we believe that combining and utilizing existing knowledge from various teachers is more effective and convenient. Furthermore, with the growing availability of large-scale pre-trained models within the community, such as hugging face, it becomes increasingly valuable to explore new ways to utilize these models and combine their strengths. This further enhances the practical value of our Hybrid Distill, and we hope our work would shed light on new directions.

## E   LIMITATION

Hybrid Distill jointly utilizes two teacher models to guide the representation learning of the student. Although exhibiting promising properties and results, the additional overhead of introducing two teachers may be a limitation. Fortunately, since the teacher model does not require gradient updates, the training cost of Hybrid Distill does not increase significantly, *i.e.*, the training time of Hybrid Distill with ViT-B backbone is around 1.2 times longer than that of using a single teacher. However, when compared to dBOT, which replaces the MAE reconstruction objective with high-level features from the teacher, Hybrid Distill enjoys shorter one-epoch training time and lower memory usage as shown in Tab. 17. This can be attributed to the fact that Hybrid Distill does not employ the resource-intensive asymmetric decoder (with 8 transformer layers) in MAE and dBOT, which, in turn, reduces the overall computational cost. Besides, when directly utilizing the available teacher models, Hybrid Distill can achieve better performance with much fewer distilling epochs, as shown in Tab. 10. Based on the above discussions, we believe that the computational cost of Hybrid Distill is acceptable. Another possible limitation is that Hybrid Distill does not improve CLIP as much as DeiT after introducing the MAE teacher, and we analyze that it may be caused by the gap between the pre-training capacities of CLIP and MAE teachers. We look forward to better MIM models that can further facilitate our work.

