# SUPPLEMENTARY MATERIALS FOR HYBRID DISTILLATION: CONNECTING MASKED AUTOENCODERS WITH CONTRASTIVE LEARNERS

## 1 COMPARED WITH MILAN

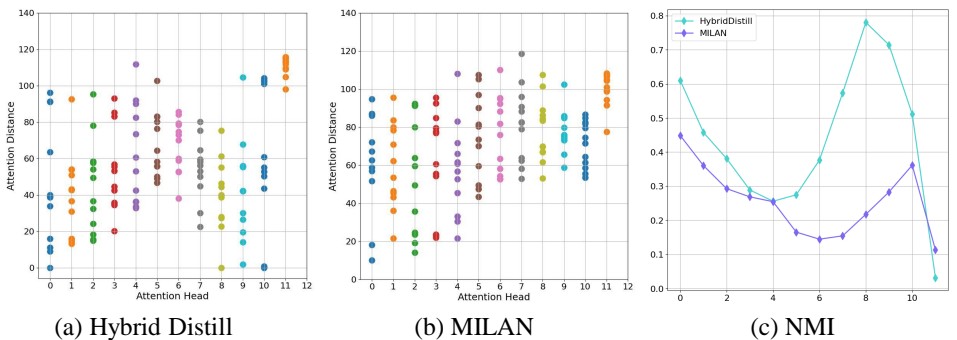

(a) Hybrid Distill      (b) MILAN      (c) NMI

Figure 1: Average attention distance of (a) Hybrid Distill and (b) MILAN and (c) their NMI.

As shown in Fig. 1 (b) and Fig. 1 (c), compared to direct distillation using an asymmetric decoder (r.f., Fig. 3 (c) and Fig. 4 (a) of our main paper), MILAN successfully improves diversity while still maintaining a certain level of discrimination. However, MILAN does not exhibit the same level of diversity gain and discrimination preservation capabilities as our Hybrid Distill (Fig. 1 (a) and Fig. 1 (c)). Therefore, it achieves lower object detection and instance segmentation performance compared to our Hybrid Distill.

## 2 MAE BEHAVIOR ON OTHER DATASET

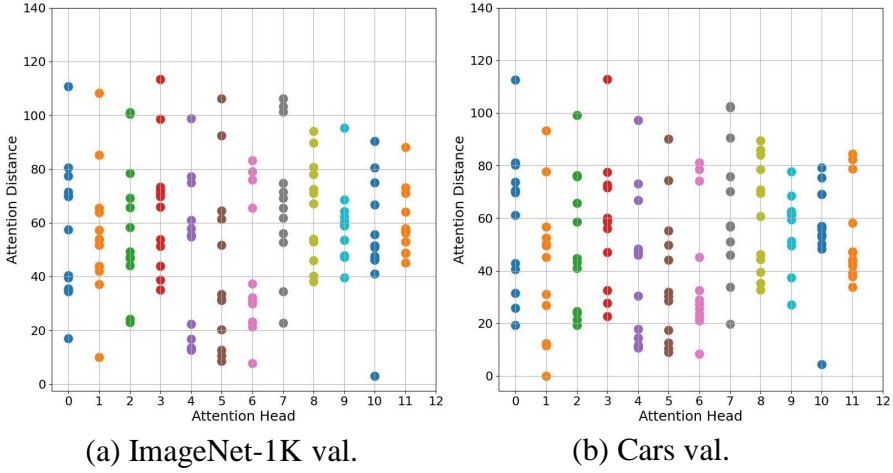

(a) ImageNet-1K val.      (b) Cars val.

Figure 2: Average attention distance of an ImageNet-1K pretrained MAE model on the (a) ImageNet-1K and (b) Cars validation sets, respectively.

Fig. 2 visualizes the average head distance of the ImageNet-1K pretrained MAE model on the ImageNet-1K and Cars validation sets, respectively. As shown, MAE performs similarly on both in-domain and out-of-domain data, *i.e.*, it maintains considerable diversity on both the ImageNet-1K and Cars validation sets. This phenomenon suggests that an MAE model pretrained on the large-scale ImageNet-1K dataset can still be beneficial for Hybrid Distill even when using data from different domains for distillation.