# OpenReview forum: "Hybrid Distillation: Connecting Masked Autoencoders with Contrastive Learners"
_ICLR.cc/2024/Conference — ICLR 2024 poster_

### Official Review · Reviewer_4NJ5 · 2023-10-31

**Soundness:** 3 good
**Presentation:** 3 good
**Contribution:** 3 good
**Rating:** 6
**Confidence:** 4

**Summary:**

The paper proposes hybrid distillation, which uses a supervised/CL teacher and a MIM teacher for a balance between discrimination and diversity. The authors use average attention distance and NMI metric to analyze the disadvantages of previous methods with respect to the above two properties, and then design hybrid distillation according to the disadvantages. Hybrid distillation achieves strong performances on various downstream tasks (classification/detection/segmentation) using ViT-B/L models.

**Strengths:**

1. The proposed hybrid distillation is simple and reasonable according to the discrimination and diversity.
2. Better performances on various downstream tasks.
3. Good writing and abundant visualization.

**Weaknesses:**

1. It seems that using two teachers for distillation really boosts the performances on downstream tasks. Will it take a longer pre-training time compared to only using CLIP as the teacher? I hope the authors can show the training time and the memory usage during distillation.
2. Previous CL-based methods outperform MAE on linear probing because they own strong discrimination abilities. Since the hybrid distillation keeps this ability, can authors show the linear probing performances on ImageNet-1k?

**Questions:**

Please refer to weakness.

---

> ### Author Response · Authors · 2023-11-16
> **To Reviewer 4NJ5**
>
> **Q1: [The training time and memory usage.]:**
>
> Please refer to Q1 for the general response, we claim that introducing additional teacher models inevitably brings more overhead, but the increase is tolerable. Besides, Hybrid Distill **enjoys shorter one-epoch training time and lower memory usage compared to dBOT[1]**. Hence, we believe that the computational cost of Hybrid Distill is acceptable compared to other distillation methods.
>
> **Q2: [Linear probing results on ImageNet-1K.]:**
>
> The linear probing performances on ImageNet-1k are shown in the table below. The subpar performance of dBOT further validates our perspective discussed in Section 2.3, which suggests that its increase in diversity comes at the expense of sacrificing discrimination. In contrast, Hybrid Distill consistently maintains outstanding linear probing performance, thus confirming the effectiveness of our approach in simultaneously promoting diversity and discrimination.
>
> **Results for linear probing. $\star$: using MAE+DeiT teachers. $\dagger$: using MAE+CLIP teachers.**
>
> | Method                   | Linear Probing |
> | :----------------------- | :------------: |
> | MAE                      |      67.8      |
> | dBOT                     |      67.9      |
> | Hybrid Distill $\star$   |      80.9      |
> | Hybrid Distill $\dagger$ |      80.1      |
>
> [1] Exploring target representations for masked autoencoders, 2022.

---

### Official Review · Reviewer_ZCMG · 2023-11-01

**Soundness:** 3 good
**Presentation:** 3 good
**Contribution:** 3 good
**Rating:** 8
**Confidence:** 3

**Summary:**

The paper studies the feature spaces and models from two different pretraining paradigms, contrastive learning  (CL) and masked image modelling (MIM). It reaffirms differences around discrimination of the produced features as well as the attention diversity and makes a number of interesting observations. Based on those, the paper proposes Hybrid Distill, a hybrid distillation method that tries to get the best of both CL and MIM. An MIM teacher is used at intermediate layers and a CL teacher at the output and this leads to models with both token diversity as well as features with strong discrimination.

**Strengths:**

* The paper studies very interesting properties and tries to understand the difference in behaviour between common learning paradigms like CL and MIM.

* The study of sec 2 is sound and with interesting observations.

* The method presented in Sec 3 is to my knowledge technically novel.

* The experimental validation seems complete and shows some gains for the proposed approach on the common setting.

**Weaknesses:**

1. Sec 2 does not discuss performance, ie the "discrimination" aspect. It is unclear if this is because the two paradigms work equally well on downstream tasks? Probably thats not exactly the case, and generalization is in the end what we care about, ie what we use these features for. Also, wrt 2.2, studies (eg Wang et al and Sariyildiz et al below as well as the papers that introduce such projectors) have shown that the size of projector/"asymmetric decoder" matters wrt to downstream task behaviour.


2. The method proposed in Sec 3 tries to get the best of both worlds by using two teachers, something that makes training harder and requires more memory.

3. More of a note than a weakness:  It is now hard to compare the relevant rows in Tables 1 and 2. Tab 1 would be more easy to understand and compare if the two sections are wrt the two different backbones (ie ViT-B on top, ViT-L on the bottom). Inside each section, a separate split should be wrt what is the teacher. The models that distill clip should be clearly separated, cause they start from a much stronger teacher. Same for the ViT-L from the authors, that distills im21k.  These results are not comparable to the rest, and the im21k-distilled row seems out of place in both tables. Please make sure to separte accordingly.

Wang, Yizhou, et al. "Revisiting the transferability of supervised pretraining: an mlp perspective." Proceedings of the IEEE/CVF Conference on Computer Vision and Pattern Recognition. 2022.

Sariyildiz, Mert Bulent, et al. "No reason for no supervision: Improved generalization in supervised models." ICLR 2023-International Conference on Learning Representations. 2023.

**Questions:**

- What do the authors mean by "s. For ViT-L, the distillation is based on both ImageNet-1K and ImageNet-21K"? It is for different models or for the same model?

- "The hyperparameter α and β are set to 1.0" - how is performance affected if the two teacher losses are not balanced like this?

---

> ### Author Response · Authors · 2023-11-16
> **To Reviewer ZCMG**
>
> **Q1: [Sec 2 does not discuss performance.]:**
>
> i)  We have demonstrated the inherent difficulty of achieving both diversity and discrimination through direct feature distillation in Section 2. Therefore, discussing the performance in Section 2 is essentially equivalent to **comparing the superiority of these two properties with respect to different downstream tasks since the student can only guarantee one of these properties, and this comparison has already been discussed in the second paragraph of Section 1 and in previous related works [1-2]**. Specifically, models with better discrimination exhibit advantages in recognition tasks, particularly in linear probing, while models with better diversity excel in dense-level tasks.
>
> ii) In Q2 ii) of the general response, we have verified that models with better discrimination indeed exhibit superior linear probing performance. However, according to previous works [2-3] and our experience, longer pre-training or distillation periods (e.g., 1600 epochs in [2-3]) are required to achieve sufficient diversity and reap dense-level downstream benefits. In Section 2, we only conduct 100 epoch distillation to compare the distillation trends, which is insufficient to fully reflect the downstream advantages of diversity. Therefore, performance comparisons at this stage have limited value. Nonetheless, our Hybrid Distill design is based on the analysis presented in Section 2, and we consider its effectiveness on dense-level tasks as reliable evidence supporting this analysis.
>
> iii)  Regarding the two papers you mentioned, they primarily discuss the impact of projectors on CL and Supervised models, whereas our focus is on their influence on the trend of distillation. As a result, they have different motivations with our studies.
>
> **Q2: [The method makes training harder and requires more memory.]:**
>
> i) Please refer to Q1 of the general response, Hybrid Distill **enjoys lower memory usage compared to MAE and dBOT**. Hence, we believe that the computational cost of Hybrid Distill is acceptable compared to other distillation methods.
>
> ii) Regarding the training difficulty, we argue that Hybrid Distill actually mitigates this issue since the required properties can be directly obtained from teacher models. This claim is supported by the results presented in Table 10, as well as the discussion provided in Section E of the appendix. Specifically, Hybrid Distill can achieve better performance with much fewer training epochs compared to dBOT.
>
> **Q3: [Hard to compare the relevant rows in Tables 1 and 2.]:**
>
> Thanks for your suggestion and we have reorganized Table 1 and Table 2 in the revised version.
>
> **Q4: [The distillation schedule when using ViT-L.]:**
>
> For ViT-L, we offer two distinct models that are derived through different distillation processes. The first model is obtained through a 100-epoch distillation process using ImageNet-1k, while the second model is derived from a 40-epoch distillation process with ImageNet-21k. We have made revisions to the description in order to improve its clarity.
>
> **Q5: [The hyperparameter α and β.]:**
>
> Please refer to Table 13 and Table 14 and the discussion in Section A.5 of the appendix, we ablate the setting of α and β and conclude that: i) Combining two teachers for hybrid distillation can lead to performance gains towards only using one teacher, regardless of the value of hyperparameters α and β. ii) Ensuring the weights of the more powerful model among the two teacher models count for better performance. iii) Setting both α and β to 1 for simplicity can already yield satisfactory results.
>
> [1] What Do Self-Supervised Vision Transformers Learn? 2023.
>
> [2] Revealing the Dark Secrets of Masked Image Modeling, 2022.
>
> [3] Masked Autoencoders Are Scalable Vision Learners, 2021.

---

> > ### Comment · Reviewer_ZCMG · 2023-11-22
> > **Thank you for your responses**
> >
> > I think my questions have been successfully covered with the responses  and keep my accept recomendation for now. Looking forward to more discussion with the other reviewers.

---

### Official Review · Reviewer_zDe2 · 2023-11-05

**Soundness:** 2 fair
**Presentation:** 3 good
**Contribution:** 3 good
**Rating:** 6
**Confidence:** 4

**Summary:**

This paper proposes a distillation or learning method for pretraining visual representation. Motivated from the differences in MAE, DeiT, CLIP models, in terms of feature diversity and attention distance, this paper proposes to distill differently for the two targets, specifically learn features from DeiT/CLIP models and attention maps from MAE methods. Results show some improvements over baselines on classification, detection, segmentation.

**Strengths:**

1. The motivation is presented quite clearly given existing studies on average attention distance and NMI.
2. The ablations are extensive and convincing that HybridDistill helps compared with vanilla MAE or DeiT.

**Weaknesses:**

1. Unfair comparison. Table 1 compares to many vanilla training methods on ImageNet-1K but Hybrid Distill exploits CLIP targets and ImageNet-21K for training. Other methods exploiting the joint objectives, better models, or larger scale data are not compared.
2. The two metrics of average attention distance and NMI are existing metrics. They provide insights to models but are still neither necessary or sufficient condition for a good representation. Existing works such as dBOT also visualized the average attention distance differences in existing methods (DeiT, DINO, MAE, etc.) and even different distillation stages and observed similar conclusion as well. Existing joint learning work with the distillation objective is producing good representation already.
3. HybridDistill is complicated and designed specifically to ViT architecture with global attention map. Therefore, it looks challenging to apply to other architectures such as Swin or MViT or CNNs.
4. Marginal performance gain compared with existing solutions in fair comparison. E.g. SdAE in Table 1 achieves 84.1, iBOT achieves 84.0, etc. with ViT-B without CLIP or ImageNet-21K, but Hybrid Distill achieving 83.7 lags behind. Similarly limited gain is observed with CLIP targets, e.g. dBOT 87.8% vs. this paper 87.6% with ViT-L ImageNet-1K CLIP targets, and Cascaded detection settings and ADE20K segmentation.
5. Runtime compared with baselines. Since Hybrid Distill requires inferencing with multiple backbones and uses less aggressive masking strategy, what is the runtime of one epoch compared with vanilla MAE?
6. The claim of using fewer epochs isn’t quite important here considering the computation used in training all the targets (including CLIP).

**Questions:**

(see weakness for details)
------------------------------
After reading all the reviews and discussions, I feel that this paper is still around borderline, and it's ok to accept it. Empirically, it's unfortunately not very useful, or to say, it's only useful in very extreme cases (when there are MAE and DEiT models, but no dBOT model, and without considering the complexity of coding, then it can be trained a bit faster). However, if it is to be accepted, these extensive studies and observations may still be interesting to some readers, e.g. Reviewer ZCMG.

---

> ### Author Response · Authors · 2023-11-16
> **To Reviewer zDe2 (1/2)**
>
> **Q1: [Unfair comparison in Table 1.]:**
>
> i) Since the main purpose of our work is to reveal the incompleteness of previous works and to verify the benefits of promoting diversity and discrimination simultaneously, **we conduct most of our experiments in a basic setting**. As shown in Table 1 and Table 9 of our appendix, we **conduct 300 epochs of distillation utilizing the ImageNet-1K dataset and ViT-B/L as backbones**. We only report the results with ImageNet-21K in the last row of Table 1 to illustrate the potential benefits of larger-scale distillation, **while all other results are reported within the basic setting for fair comparisons**.
>
> ii) Besides the CLIP+MAE targets you mentioned, **Hybrid Distill also provides results of using DeiT+MAE targets** and both results showcase consistent benefits towards teacher models and distillation baselines on dense-level detection and segmentation tasks, illustrating the effectiveness of our Hybrid Distill. We have reorganized Table 1 to make the relevant comparisons easy to understand.
>
> **Q2: [Average attention distance and NMI are existing metrics.]:**
>
> We agree that average attention distance and NMI do not directly indicate good representation, as they are merely tools used to assess model behavior. However, **we argue that these two metrics have been inadequately interpreted in previous works**. In fact, our work is dedicated to **solving the limitations of the distillation works you mentioned, including dBOT [1]**. We present evidence that the diversity brought by dBOT is derived from its asymmetric decoder design (Section 2.2), while the asymmetric decoder design de facto hinders discrimination (Section 2.3). **This loss in discrimination also explains the phenomenon you mentioned, where models with different teachers (DeiT, DINO, MAE, etc.) exhibit similar behavior in dBOT after multi-stage distillation**. In summary, in Section 2, we have confirmed that **existing works are still far from achieving good representation from analyzing the diversity and discrimination perspectives**.
>
> **Q3: [Apply to other architectures such as Swin or MViT or CNNs.]:**
>
> The core of our Hybrid Distill is to utilize MAE and Supervised/CL teachers to ensure both diversity and discrimination. Since **these two kinds of teachers have been confirmed to demonstrate similar behavior in hierarchical transformer structures (e.g., Swin) [2]** when compared to the plain ViT structure, we believe that our designs can be effectively adapted to these hierarchical structures as well. However, our analysis relies on the available pretrained models for efficiency, and the challenge lies in that previous methods have not provided pre-training weights (e.g., CLIP) or baseline results (e.g., [1,3-5]) for these structures. Therefore, further comprehensive validation will be conducted in the future, in parallel with the research progress of the community.

---

> > ### Author Response · Authors · 2023-11-16
> > **To Reviewer zDe2 (2/2)**
> >
> > **Q4: [Marginal performance gain compared with existing solutions.]:**
> >
> > i)  **Hybrid Distill achieves noticeable gains in the dense-level downstream for its diversity advantage**. Specifically, when distilled with DeiT+MAE teachers, Hybrid Distill results in a growth of 1.4 $AP_{box}$ and 1.2 $AP_{mask}$ compared to SdAE (r.f., Table 1). In comparison to iBOT, Hybrid Distill demonstrates gains of 1.7 $AP_{box}$ and 1.3 $AP_{mask}$ when using Cascaded Mask-RCNN for downstream adaption  (r.f., Table 10 of the appendix). These improvements are in line with the analysis in previous works [2,6], which indicates that diversity is mainly beneficial for dense-level downstream. While for classification tasks, performance on ImageNet is affected by the saturation of the DeiT teacher. Hence, we focus more on its ability to generalize to other datasets (r.f., Table 2). Specifically, Hybrid Distill achieves 1.4% and 1.2% average accuracy gains towards DeiT and MAE teachers, respectively.
> >
> > ii)  **The performance comparison you mentioned between dBOT and Hybrid Distill is not conducted under fair conditions**, as Hybrid Distill only uses a basic 300-epoch distillation setting while dBOT distills the model for 1600 epochs.  Nevertheless, we believe that the fact that **300 epoch** Hybrid Distill can achieve comparable or even better performance (r.f., Table 10 of our appendix) compared to **1600 epochs** dBOT already demonstrates its effectiveness. Besides, we have provided 300 epoch dBOT distillation results in Table 9 of the appendix, and also supplement the 600 epoch dBOT distillation results in the table below. **In these fairer comparisons, the advantages of our Hybrid Distill are more pronounced**.
> >
> >
> > | Method         |   Teacher    |  Epoch  | $AP_{box}$ | $AP_{mask}$ |
> > | :------------- | :----------: | :-----: | ---------: | ----------: |
> > | dBOT           |     DeiT     | 300/600 |  47.5/47.7 |   41.9/42.0 |
> > | dBOT           |     MAE      | 300/600 |  49.3/49.6 |   43.5/43.7 |
> > | dBOT           |     CLIP     | 300/600 |  49.5/49.8 |   43.5/43.8 |
> > | Hybrid Distill | MAE and DeiT |   300   |       50.3 |        44.2 |
> > | Hybrid Distill | MAE and CLIP |   300   |       50.6 |        44.4 |
> >
> > **Q5: [Runtime compared with baselines. ]:**
> >
> > Please refer to Q1 of the general response, Hybrid Distill **enjoys shorter one-epoch training time compared to dBOT and lower memory usage compared to MAE and dBOT**. Hence, we believe that the computational cost of Hybrid Distill is acceptable compared to other distillation methods.
> >
> > **Q6: [The claim of using fewer epochs isn’t quite important.]:**
> >
> > i) Our claim about "fewer epochs and better performance" is made in comparison to other distillation methods, while **the additional costs for training the targets are inherent to all these distillation-based methods**.
> >
> > ii) In fact, since there is a growing availability of off-the-shelf pretrained models in the community, such as hugging face, **it has become a more promising trend to consider how to efficiently utilize these pre-trained models**, and distillation is an efficient strategy for reusing these off-the-shelf model with better representation ability. Therefore, when directly leveraging these off-the-shelf models, the cost of pretraining is generally not considered in the process of distillation.
> >
> > iii) Our objective in comparing efficiency is to highlight the superior ability of Hybrid Distill in effectively utilizing off-the-shelf models compared to other distillation methods, and we believe this efficiency advantage will contribute to the community.
> >
> > [1] Exploring target representations for masked autoencoders, 2022.
> >
> > [2] Revealing the Dark Secrets of Masked Image Modeling, 2022.
> >
> > [3] Mvp: Multimodality-guided visual pre-training, 2022.
> >
> > [4] BEiT v2: Masked Image Modeling with Vector-Quantized Visual Tokenizers, 2022.
> >
> > [5] EVA: Exploring the Limits of Masked Visual Representation Learning at Scale, 2022.
> >
> > [6] What Do Self-Supervised Vision Transformers Learn? 2023.

---

> > > ### Comment · Reviewer_zDe2 · 2023-11-22
> > > **Follow up discussion/comment on Q4 and Q6.**
> > >
> > > Thanks for the detailed response. Here are more discussions/comments related to Q4 and Q6:
> > >
> > > In practice, there are two cases where the proposed distillation method may be useful for practitioners:
> > >
> > > (1) When someone has in-house data likely from a different domain and wants to learn representation on such custom data (assuming the CLIP model still applies to this domain): In this case, reproducing HybridDistill on custom data will require 1600 epochs of the original MAE training plus the 300 epochs of distillation (1900 in total), which will put HybridDistill at a disadvantage in both cost and performance even compared with 1600 epochs of distillation in dBOT (Table 10). In this case, the fewer epochs of HybridDistill or the memory-saving property of HybridDistill won't matter because a custom MAE stage is needed on the custom data. In this case, HybridDistill is more costly.
> > >
> > > (2) When someone is simply interested in using a general visual representation on ImageNet and wants to exploit anything that exists online in a model zoo or hugging face, then the most reasonable action is probably to download the 1600-epoch dBOT-CLIP (i.e. 0 epoch of custom training needed) instead of downloading the 1600-epoch MAE and then training 300-epoch of HybridDistill. In this case, dBOT-CLIP provides better representation.

---

> ### Author Response · Authors · 2023-11-23
> **Reply to further comment**
>
> Thanks for your detailed comment.
>
> **Q1: [Using in-house data from a different domain.]:**
>
> i) Compared with directly distilling on custom data, we think that distilling on a large-scale, general dataset and then fine-tuning on custom datasets (typically with small-scale supervised data or weakly supervised data) may be more applicable for its effectiveness in acquiring high-quality representation and preventing potential overfitting. Additionally, models obtained through large-scale distillation have the capability to quickly adapt to different custom data, as downstream tasks only require further fine-tuning on relatively small-scale datasets. While **for large-scale distillation itself**, as mentioned in our response to Q5 and Q6, **Hybrid Distill has efficiency advantages** towards dBOT.
>
>
> ii) **Even in cases where distillation needs to be performed on custom data, the MAE model obtained through large-scale pre-training can still serve as a good teacher**. As demonstrated in Section 2 of our updated supplementary materials, **the unsupervised training paradigm of the MAE teacher allows it to maintain considerable diversity even when applied to data from different domains**. Therefore, we believe that utilizing this model to provide diversity for Hybrid Distill is still effective and the efficiency advantages of our Hybrid Distill can still be maintained.
>
>
> **Q2: [Using a general visual representation on ImageNet.]:**
>
> i) In our response to Q6, we have emphasized that Hybrid Distill is more efficient than dBOT in utilizing the off-the-shelf teacher models (e.g., DeiT, CLIP, and MAE) for distillation. **While the dBOT models**, obtained through distillation, do not belong to these base teacher models and they **should be compared with our Hybrid Distill models (also obtained through distillation) for fair comparisons**. We have further improved our revised version's description about 'fewer epochs' to improve clarity.
>
> ii) Note that  **dBOT also requires custom training, specifically through the process of distillation**. The models provided by dBOT to the community have already undergone **1600 epochs of custom training** under the guidance of a pre-trained teacher, but this does not mean that custom training is unnecessary for dBOT.  The advantages of our Hybrid Distill lie in its distillation strategy, which results in a significantly lower custom training cost compared to dBOT. Therefore, **we can also provide the community with performance-enhanced models that require less custom training cost (e.g., 300 epochs, or 500 epochs for further benefits) by leveraging our Hybrid Distill paradigm**, and releasing such models is in our plan.

---

### Official Review · Reviewer_mHak · 2023-11-08

**Soundness:** 3 good
**Presentation:** 2 fair
**Contribution:** 2 fair
**Rating:** 6
**Confidence:** 4

**Summary:**

After discussing the "diversity" and "discrimination" in the features of a distilled Transformer, this paper proposed a hybrid distillation framework that simultaneously distills knowledge from two distinctively different pre-trained teacher. Specifically, HybridDistill designs the distilling target and location from a pre-trained MIM model and a supervised/CL model. A progressive redundant token masking strategy is also proposed for reducing the distilling costs.

**Strengths:**

- The paper provides many results and discussion on comparing the pre-head attention distance with different types of teacher model, which could be a helpful resource for the community.
- The paper provides through ablation studies on each component of the proposed method.

**Weaknesses:**

- My biggest concerns are on the validity of the motivation to encourage both "diversity" and "discrimination" in the Transformer.
  - About "diversity":
    - This submission claims that "diversity" ```means that the model pays attention to both local and global information and can achieve more evenly distributed representations```. What is ```evenly distributed representations'''? Specifically, which does it mean by "distributed representation"?
    - My understanding is that the "diversity" is quantitively measured by both average attention distance and NMI. I am not sure if "diversity" is a good terminology in this case, which oversimplified many necessary context. I suggest, for example, "different per-head attention distance", to more accurately describe the discussed property. For simplicity, I will still use "diversity" in my review.
    - I am **especially** not fully convinced that “diversity” itself is something clearly worth to pursue in a pre-trained Transformer. Yes, "diversity" is evaluated in several previous works, but often as a side information/evidence to better understand the self-attentions. However, it is not convincing that “diversity” in the last layers is something to specially encouraging. For a starter, what *should* be *ideal* per-head attention distance in each layer already requires serious discussion. It is hard to believe that focusing on long-term/global information in the last layers is unquestionably bad. If the authors argue that "diversity" in last layers may help dense-level tasks, while, then why not simply make better usage of multi-layer features in these dense tasks instead of forcing last layer features to be "diverse"?
  - About "discrimination":
    - This is a question: what metric is used to quantitively evaluate "discrimination" in the experiments?

- Questions on results:
  - Why MAE's COCO results in the main Table (Table 1) are **lower** than the results reported in MAE's original paper? For example, in MAE, a ViT-B is reported to get 50.3 APbox and 44.9 APmask, but 48.4 APbox and 42.6 APmask in this submission? Please highlights the experimental setting differences to MAE and other compared baselines in the paper if this is the reason. However, if the setting difference is the reason for performance difference on COCO, then why ADE20K results are matched?
  - If directly distilling CLIP's feature harms dense tasks for its inadequate "diversity" in self-attention distance, then why results reported in MILAN outperforms the proposed method on COCO detection/instance segmentation and ADE20K semantic segmentation? I think MILAN directly predictions CLIP's features. Also, MILAN is not cited and compared in the submission.

[MILAN] Hou et al, MILAN: Masked Image Pretraining on Language Assisted Representation.


-----


Post-Rebuttal Comment: I thank the authors for the response, which partially addressed my concerns on the naming/motivation of "diversity" vs. discrimination, and the results on COCO compared to previous methods. I adjusted my recommendation accordingly.

But I still want to highlight: Although previous works [3-5] adopting shorter schedules seem common, I believe that fine-tuning for a long schedule to see the real performance given enough training is very important to illustrate the effectiveness of the proposed methods. Sometimes the pre-trained models only show effectiveness with shorter schedule, and the gains diminish after longer training. Longer schedule provides a lens for us to better understand the proposed method.

**Questions:**

Discussions/addressing the questions in **bold** in Weakness will be very helpful. Thanks!

---

> ### Author Response · Authors · 2023-11-16
> **To Reviewer mHak (1/2)**
>
> **Q1: [Questions about diversity and discrimination.]:**
>
> We actually share a similar viewpoint as you, i.e., we acknowledge **the importance of preserving long-term information (i.e., discrimination) in the last layers** and **do not believe that diversity is the only property to pursue**. The fundamental principle of our Hybrid Distill is thus to **retain this long-term information while simultaneously promoting diversity**. In the following, we talk about your concerns about diversity and discrimination point by point.
>
> **[The definition of diversity.]**
>
> The term 'evenly distributed representations' is derived from [1], indicating that the representations are acquired through more evenly distributed attention across different attention distances. Your interpretation is also accurate, as diversity does represent the existence of 'different, or diverse, per-head attention distances'. In the revised version, we have adjusted the description for clarity.
>
> **[Is diversity itself clearly worth to pursue?]**
>
> i) As mentioned at the beginning, we also recognize the importance of preserving long-term information in the last layers as well as the limitation of judging the model with only diversity like previous works [1-2]. We emphasize that **increasing diversity should not be at the expense of sacrificing discrimination (i.e., long-term/global information in the last layer)**. This notion forms the core of our discussion in Section 2 and serves as the foundation for the development of Hybrid Distill in Section 3.
>
> ii)  We acknowledge the significance of long-term information, **but the problem lies in the fact that the model uses too many transformer layers to represent homogeneous long-term information**. We introduce diversity to address this homogeneity, not to replace this information entirely. It is worth noting that although this problem is more apparent in the later layers, the trend of homogeneity can already be observed in earlier layers of CL/Supervised teachers. Therefore, relying solely on multi-layer features is not an essential solution to this problem. In fact, **we have already incorporated multi-layer features in our downstream tests for detection and segmentation** (r.f., Section C of the appendix), and the clear advantages of our Hybrid Distill over direct distillation (r.f., Table 1 of our paper) in these downstream provide support for our claim.
>
> **[The quantitively evaluation of discrimination.]**
>
> Please refer to Q2 of the general response, we utilize average attention distance and NMI to judge both diversity and discrimination. Besides, we have included additional linear probing results  in the revised version, and the results align with the analysis of discrimination presented in Section 2.
>
> **Q2: [MAE results. ]:**
>
> Sorry for the confusion, the difference between reported performance and original values in the MAE paper is attributed to **variations in downstream settings**.  We follow the settings of recent works  [3-5] (which are also widely adopted currently) for employing pretrained models in downstream tasks of COCO detection and ADE20K segmentation. For COCO detection:
>
> i) **The MAE paper finetunes the pretrained model for 100 epochs**, which is time-consuming and not a common downstream setting. Besides, MAE does not release the official codes for COCO detection and some fine-tuning details are missing. Therefore, we follow the common practice in [3-5] that **employs a 1x schedule (12 epochs)** for fine-tuning, and our reported MAE results align with these papers (48.4 $AP_{box}$ and 42.6 $AP_{mask}$).
>
> ii) In Table 10 of our appendix, **we also provide another set of COCO detection results using Cascaded Mask-RCNN for comprehensiveness**. According to [2], the reported MAE result under this setting is 50.6 $AP_{box}$, while our Hybrid Distill achieves 53.0 $AP_{box}$, which is 2.4 points higher than MAE. This clearly demonstrates the advantages of our method compared to MAE across different settings.

---

> ### Author Response · Authors · 2023-11-16
> **To Reviewer mHak (2/2)**
>
> **Q3: [MILAN is not cited and compared in the submission.]:**
>
> i) **MILAN does not directly distill features**. Instead, it introduces a prompting decoder design that involves freezing the unmasked tokens. These tokens are then used as keys and values to interact with learnable mask tokens in the decoder. In our updated supplementary materials, we have verified that this design can alleviate the discrimination loss caused by the asymmetric decoder, resulting in improved performance. Therefore, the reason for its performance gain compared to direct distillation can be explained by the metrics in our paper, i.e.,  it successfully improves diversity while still maintaining a certain level of discrimination. We have cited this paper in the revised version.
>
> ii) MILAN follows the downstream setting of MAE and also does not disclose its details. Therefore, **we re-evaluate the performance of MILAN under our 1x downstream setting for a fair comparison, and our method outperforms MILAN by 0.9 $AP_{box}$ and 0.7 $AP_{mask}$**, as demonstrated in the table below. Evidence for this improvement can be found in the supplementary materials, i.e., MILAN does not exhibit the same level of diversity gain and discrimination preservation capabilities as our Hybrid Distill.
>
> **Results compared with MILAN**
>
> | Method         |   Teacher    | $AP_{box}$ | $AP_{mask}$ |
> | :------------- | :----------: | :--------: | ----------: |
> | MILAN          |     CLIP     |    49.7    |        43.7 |
> | Hybrid Distill | MAE and DeiT |    50.3    |        44.2 |
> | Hybrid Distill | MAE and CLIP |    50.6    |        44.4 |
>
>
> [1] Contrastive Learning Rivals Masked Image Modeling in Fine-tuning via Feature Distillation, 2022.
>
> [2] Exploring target representations for masked autoencoders, 2022.
>
> [3] Context Autoencoder for Self-Supervised Representation Learning, 2022.
>
> [4] Sdae: Self-distillated Masked Autoencoder, 2022.
>
> [5] Progressively Compressed Auto-Encoder for Self-Supervised Representation Learning, 2023.

---

### Author Response · Authors · 2023-11-16
**General Response**

We sincerely appreciate all reviewers for their efforts. we first give general responses for some common issues.

**Q1: [Runtime and memory usage compared with baselines.]**

i) Introducing an additional teacher model inevitably brings more overhead. However, as discussed in Section E of the appendix, since the teacher does not require gradient updates, the training cost does not increase significantly, e.g., the training time of Hybrid Distill is **around 1.2 times longer** than that of directly distilling with a single teacher when using ViT-B as the backbone.

ii) In addition, we show the one-epoch training time and memory usage of dBOT [1] and Hybrid Distill, as well as the original training cost of MAE in the table below. **Compared with the typical distillation method dBOT, which replaces the MAE reconstruction objective with high-level features from the teacher, Hybrid Distill enjoys shorter training time and lower memory usage**.  This can be attributed to the fact that Hybrid Distill does not employ the resource-intensive asymmetric decoder (with 8 transformer layers) as in MAE and dBOT, thus reducing the overall computational cost.

iii) Furthermore, as mentioned in Section E of the appendix, **Hybrid Distll can achieve better performance with much fewer training epochs compared to dBOT**, e.g., 300 epoch Hybrid Distill achieves 53.0 $AP_{box}$ with Cascade Mask-RCNN when using MAE+DeiT teachers, outperforming 1600 epoch dBOT-DeiT (52.5 $AP_{box}$) and dBOT-MAE (52.7 $AP_{box}$).  Based on the above discussions, we believe that the computational cost of Hybrid Distill is acceptable. We have further supplemented the point ii) in Section E in the revised version.

**One-epoch training time and memory usage. Test based on ViT-B with 128 per-GPU batch size on 8 v100.**

| Method         | One-epoch training time | GPU memory |
| :------------- | :---------------------: | :--------: |
| MAE            |         9min15s         |   17807M   |
| dBOT           |        12min42s         |   18953M   |
| Hybrid Distill |        11min12s         |   9503M    |

**Q2: [The measurement of discrimination in Section 2.]** -

i) Average attention distance and NMI are metrics that indicate both diversity and discrimination. Previous works [1-4] have shown that models with better discrimination, such as Supervised/CL pre-trained models, tend to exhibit long-range attention distance in the last few layers. Additionally, the low NMI value further suggests that the generation of this long-distance attention is primarily attributed to all queries focusing on similar tokens, and we show evidence in Figure 6 that these similar tokens tend to correspond to the main subject of the input image. Consequently,  **we define the model with long-range attention distance and low NMI value at the last layer as having discrimination**.

ii) As shown in the table below, we have included additional linear probing results to judge the discrimination. These findings are consistent with the analysis presented in Section 2, indicating that the asymmetric decoder negatively impacts the discrimination (Section 2.3) and the behavior of mask feature reconstruction is similar to that of directly distilling features (Section 2.4).  These additional results provide further support to our analysis of discrimination, and we have added them in the revised version.

**Results for Linear Probing when using the DeiT Teacher. The distillation configurations are the same as in Section 2.**

| Distillation configuration              | Linear Probing |
| :-------------------------------------- | :------------: |
| No decoder (Fig. 3(a))                  |      81.4      |
| Linear projection (Fig. 3(b))           |      81.7      |
| Asymmetric decoder(Fig. 3(c))           |      66.2      |
| Mask feature reconstruction (Fig. 3(d)) |      79.5      |

[1] Exploring target representations for masked autoencoders, 2022.

[2] Contrastive Learning Rivals Masked Image Modeling in Fine-tuning via Feature Distillation, 2022.

[3] Revealing the Dark Secrets of Masked Image Modeling, 2022.

[4] What Do Self-Supervised Vision Transformers Learn? 2023.

---

### Meta-Review · Area_Chair_gxj8 · 2023-12-13

**Metareview:**

This paper introduces a hybrid distillation framework leveraging knowledge from both a MIM teacher model and a Contrastive Learning teacher model. A progressive redundant token masking strategy is also introduced to reduce the distilling costs. Extensive empirical results across multiple downstream tasks (including classification, detection, and segmentation) are provided to support the effectiveness of the proposed method.

Overall, the reviewers find this paper interesting to read, and appreciate the simplicity and the effectiveness of the proposed methods. But meanwhile, several major concerns are raised: 1) further conceptual clarity and validation of 'diversity' and 'discrimination' is needed; 2) some empirical comparisons are possibly unfair and need further clarifications; 3) more detailed computational analysis is needed.

The authors provide a comprehensive rebuttal to the above questions, successfully addressing most major concerns. All reviewers unanimously agree to accept this paper. In the final version, please include all these clarifications to enhance the quality of this paper.

**Justification For Why Not Higher Score:**

As mentioned by Reviewer zDe2, this framework has certain constraints and therefore may not be very practically useful in its current version.

**Justification For Why Not Lower Score:**

As acknowledged by reviewers, the extensive studies and observations provided in this paper will be interesting to the community.

---

### Decision · Program_Chairs · 2024-01-16

Accept (poster)